# Longitudinal healthcare use after pediatric brain injury: A population-based birth cohort study

Vincy Chan[1,2,3,4]*, Clarissa Serafine Wirianto[1], Robert Balogh[2], Michael David Escobar[5]

**1** KITE Research Institute, Toronto Rehabilitation Institute-University Health Network, Toronto, Canada, **2** Faculty of Health Sciences, Ontario Tech University, Toronto, Canada, **3** Institute of Health Policy, Management and Evaluation, University of Toronto, Toronto, Canada, **4** Rehabilitation Sciences Institute, Temerty Faculty of Medicine, University of Toronto, Toronto, Canada, **5** Dalla Lana School of Public Health, University of Toronto, Toronto, Canada

\* vincy.chan@uhn.ca

## Abstract

### Background

Traumatic brain injury is a chronic disease with lifelong consequences. In children, it can affect developmental milestones. Longitudinal data on brain injury and long-term healthcare use is limited, with lack of clarity on social determinants of health and its effects on healthcare use. This study explores rates of healthcare use, from birth, and up to 10 years after a childhood traumatic brain injury-related healthcare visit.

### Methods and findings

This study uses a population-based birth cohort of individuals born between April 1, 2002 and March 31, 2020 from Ontario, Canada. A case cohort (TBI cohort) was created using a sample of individuals who had at least one traumatic brain injury-related healthcare visit between the ages of 0 and 4 years, inclusive (n = 26,988). Controls were generated from a sample of individuals who did not have any traumatic brain injury-related healthcare visit during the study period (n = 193,253 for emergency department visits and hospitalizations, and n = 19,313 for primary care physician visits). The primary outcome is rates of primary care physician visits, emergency department visits, and hospitalizations for each year prior to and up to 10 years after the index traumatic brain injury-related healthcare visit, calculated using standard life table methods. Rates and 95% confidence intervals were further calculated and stratified by rurality of residence, and the following Ontario Marginalization Index metrics: neighbourhood income quintile and neighbourhood racialized and newcomer populations.

Rates of healthcare use remained consistently higher in the TBI cohort compared to controls both prior to and after the index TBI-related healthcare visit. Rates also varied across social determinants of health. Overall, rates were higher in males compared to females across all healthcare settings. Rates of primary care physician visits were higher among those living in urban (vs. rural) settings. However, rates of emergency department visits

**Data availability statement:** The dataset from this study is held securely in coded form at ICES. While legal data sharing agreements between ICES and data providers (e.g., healthcare organizations and government) prohibit ICES from making the dataset publicly available, access may be granted to those who meet pre-specified criteria for confidential access, available at www.ices.on.ca/DAS (email: das@ices.on.ca). The full dataset creation plan and underlying analytic code are available from ICES upon request.

**Funding:** Research reported in this publication was supported by the Eunice Kennedy Shriver National Institute of Child Health & Human Development of the National Institutes of Health under Award Number R03HD104206. The funders had no role in study design, data collection and analysis, decision to publish, or preparation of the manuscript. The content is solely the responsibility of the authors and does not necessarily represent the views of the National Institutes of Health. URL: https://www.nichd.nih.gov/grants-contracts.

**Competing interests:** The authors have declared that no competing interests exist.

were higher among those living in rural (vs. urban) settings. Rates of emergency department visits and hospitalizations were higher among those living in the lowest (vs. highest) income quintile neighbourhoods. Rates of primary care physician visits were higher among those living in areas with the most (vs. least) racialized and newcomer populations. However, rates of emergency department and hospitalizations were higher among those living in areas with the least (vs. most) racialized and newcomer populations. This study is limited to change in rates of healthcare use over time and does not quantify the magnitude of these changes.

## Conclusions

Research on longitudinal healthcare use is needed to explore the causes of sustained and increased healthcare use post-injury, to inform opportunities for targeted health and social care interventions. Findings also suggest that a lifespan perspective is critical to understand how early life events can impact post-injury outcome.

## Introduction

Traumatic brain injury (TBI) is a leading cause of death and disability worldwide. In 2016, there were over 27 million new cases of TBI globally, of which 47% were among children and adolescence [1]. TBIs are of particular concern among children because consequences of a TBI sustained during childhood may not be immediately apparent, with problems only manifesting years after injury when children are unable to meet developmental milestones [2]. It is also well-established that TBI is a chronic disease that impacts long-term health and quality of life [3,4]. TBIs are associated with secondary health conditions such as headaches, seizures, and endocrine dysfunction [5–7], physical and cognitive difficulties such as poor memory, motor control, and executive functioning [8–10], and social and emotional challenges such as anxiety and difficulties in language and communication [11,12]. These challenges impact society as a whole in the form of increased healthcare utilization. As such, longitudinal data that follow children are critical to identifying long-term effects on society.

To date, longitudinal data on healthcare use after TBI are scarce, a limitation recognized internationally [13]. Most data on long-term health service use after TBI are limited to the years immediately post-injury even though secondary health conditions or impairments requiring healthcare may persist or may not manifest until years after TBI. There is even less longitudinal data across social determinants of health (SDoH) despite research establishing that the health and healthcare use of individuals with TBI are strongly influenced by the consequences of TBI and intersecting SDoH [14,15]. Finally, existing birth cohorts that follow children with TBI concern those born in the 1960s to 1980s [16,17]. While these data sources continue to provide much needed evidence on long-term outcomes after TBI, the experiences of these children may differ significantly from those of children who sustained a TBI in recent decades due to the increase in research on TBI and development of clinical and return to work/play guidelines.

This study addresses the above gaps in knowledge and research by exploring the rates of healthcare use among individuals who sustained a TBI that required medical attention in early childhood. Early childhood is defined in this study as those who had at least one TBI-related healthcare visit to the primary care physician, emergency department (ED), or hospital between the ages of 0 and 4 years, inclusive. Specifically, this study reports data from a population-based birth cohort of individuals born in publicly funded hospitals in Ontario,

Canada, between April 1, 2002 and March 31, 2020. Rates of primary care physician visits, ED visits, and hospitalizations from birth and up to 10 years post-injury were determined across sex, age at the time of incident TBI-related healthcare visit, neighbourhood income quintile, rurality of residence, and neighbourhood racialized and newcomer populations characteristic (select SDoH variables available in this birth cohort). Findings from this study hold the potential to catalyze in-depth research on long-term healthcare use post-injury that can inform opportunities for targeted health and social care interventions.

## Methods

Research ethics approval was obtained from the University Health Network. Informed consent was not obtained because only de-identified data were received from ICES for this study. De-identified data were accessed by the research team on July 30, 2021.

### Data source

The population-based birth cohort was created using health administrative data from Ontario, Canada [18]. All live births in Ontario hospitals from April 1, 2002 to March 31, 2020 were identified and linked to the following health administrative datasets using a de-identified patient identifier: (a) Ontario Health Insurance Plan (OHIP) for primary care physician visits, (b) National Ambulatory Care Reporting System (NACRS) for ED visits, and (c) Discharge Abstract Database (DAD) and the Ontario Mental Health Reporting System (OMHRS) for hospitalizations.

### Sample

Individuals who had at least one TBI-related healthcare visit to the primary care physician, ED, or hospital between the age of 0 and 4 years, inclusive (hereafter referred to as 'TBI cohort') were identified using OHIP-specific or International Classification of Diseases Version 10 (ICD-10) diagnosis codes in the OHIP and NACRS, DAD, and OMHRS databases, respectively (n = 269,295). Controls were individuals who did not have any TBI-related healthcare visits during the study period. Due to computational limitations, a random 10% sample of the TBI cohort was taken for analyses exploring rates of primary physician visits (n = 26,988). Similarly, due to computational limitations, a random 10% sample of controls was taken for analyses exploring rates of ED and hospitalizations (n = 193,253) and a further random 10% sample was taken for analyses exploring the rates of primary care physician visits (n = 19,313). Random samples were computer-generated using a uniform distribution. Supplementary 1 presents a flow diagram of the sample (S1 Fig), lists the OHIP and ICD-10 codes for identifying TBI-related healthcare visits (S1 Table), and describes the computational limitations (S1 Text).

### Variables

The outcome variables were the rates of healthcare use – i.e., (a) primary care physician visits, (b) ED visits, and (c) hospitalizations, in each year before and after the index TBI-related healthcare visit, up to 10 years after the index TBI-related healthcare visit.

Rates of healthcare use across: (a) sex, (b) rurality of residence, determined by the location of the individual's postal code of residence at the time of TBI, (c) neighbourhood income quintile, based on the individual's postal code at time of index TBI-related healthcare visit, and (d) neighbourhood racialized and newcomer populations characteristic (measuring recent immigrants and people belonging to a 'visible minority' group, based on the Ontario Marginalization Index, and determined by linking the individuals' postal code at time of index

TBI-related healthcare visit to the Ontario Marginalization Index database), were identified to explore similarities and differences across select SDoH variables available in the birth cohort.

## Statistical analyses

All statistical analyses were conducted using R version 3.6.3 and parallel processing with 10 cores.

Two-tailed weighted Chi-squared tests were performed for group comparison statistics. Results were considered significant at p < 0.001.

Rates of healthcare use were calculated for each healthcare setting (i.e., primary care physician, ED, and hospital) by age at the time of the incident TBI-related healthcare visit (i.e., ages 0, 1, 2, 3, and 4) for the TBI cohort and, for the control cohort, based on whether they were in the population at the corresponding age of patients in the TBI cohort. As such, control individuals may be included more than once across multiple ages. Standard life table methods, accounting for person-years at risk, were used such that the number of visits to a healthcare setting was divided by the total number of person-years still in the population:

$$Rate_i = \frac{Total\ visits_i}{Total\ person\ years_i}. \tag{1}$$

where $i$ = time in years.

Similarly, rates of healthcare use by SDoH were calculated per healthcare setting and by age at the time of incident TBI-related healthcare visit for the TBI cohort and, for the control cohort, by whether they were in the population at the corresponding age of patients in the TBI cohort. Due to large sample sizes, all confidence intervals were within 0.02 units of the rate and left out of the results for succinctness.

$$CI = Rate_i \pm Z_{\alpha/2} \sqrt{\frac{Total\ visits_i}{Total\ person\ years_i^2}} \tag{2}$$

## Results

In the TBI cohort, 56.9% were males and, at the time of index TBI-related healthcare visit, 7.9% lived in rural neighbourhoods, 20.8% lived in the lowest income quintile neighbourhoods, and 34.3% lived in areas with the most racialized and newcomer populations. Over 95.0% of index TBI-related healthcare visits occurred in primary care physician offices. Among controls, 49.9% were males and, at the time of birth, 9.4% lived in rural neighbourhoods, 22.4% lived in the lowest income quintile neighbourhoods, and 34.8% lived in areas with the most racialized and newcomer populations. Table 1 presents characteristics of the TBI and control cohorts.

## Rates of primary care physician visits

Overall, rates of primary care physician visits for the TBI cohort ranged from 398.4 per 100 person-years to 1170.5 per 100 person-years and for the control cohort, ranged from 304.2 to 1437.0 per 100 person-years (Fig 1A). Rates for both the TBI cohort and controls were highest in the first two years of life regardless of age at the index TBI-related healthcare visit. Compared to controls, the rate of primary care physician visits for the TBI cohort was significantly lower during the first two years of life, after which they were significantly higher. The rate of visit among the TBI cohort was on average 13.2% (age 0) to 24.9% (age 4) higher than that of controls over the 10 years post-injury, when stratified by age. Overall, the rate of primary

**Table 1. Sociodemographic information on sample by social determinants of health variables.**

| | TBI Cohort | | Control Cohort | | Chi-squared p-value |
|---|---|---|---|---|---|
| | N | % | N | % | |
| Total | 269295 | 100 | 193253 | 100 | NA |
| **Age at Incident TBI-Related Healthcare Visit** | | | | | |
| 0 | 82991 | 30.8 | NA | NA | NA |
| 1 | 75716 | 28.1 | NA | NA | |
| 2 | 47312 | 17.6 | NA | NA | |
| 3 | 33702 | 12.5 | NA | NA | |
| 4 | 29574 | 11.0 | NA | NA | |
| **Sex** | | | | | |
| Male | 153311 | 56.9 | 96504 | 49.9 | <0.001 |
| Female | 115984 | 43.1 | 96749 | 50.1 | |
| **Rurality** | | | | | |
| Rural | 21208 | 7.9 | 18095 | 9.4 | <0.001 |
| Urban | 248087 | 92.1 | 175158 | 90.6 | |
| **Income Quintile** | | | | | |
| 1 - Lowest | 55944 | 20.8 | 43308 | 22.4 | <0.001 |
| 2 | 52536 | 19.5 | 39161 | 20.3 | |
| 3 | 54748 | 20.3 | 39691 | 20.5 | |
| 4 | 58663 | 21.8 | 39506 | 20.4 | |
| 5 - Highest | 47404 | 17.6 | 31587 | 16.3 | |
| **Racialized and Newcomer Populations** | | | | | |
| 1 - Least | 31117 | 11.6 | 25249 | 13.1 | <0.001 |
| 2 | 38013 | 14.1 | 27932 | 14.5 | |
| 3 | 47042 | 17.5 | 32105 | 16.6 | |
| 4 | 60637 | 22.5 | 40811 | 21.1 | |
| 5 - Most | 92486 | 34.3 | 67156 | 34.8 | |

[a]Controls at later ages are a subset from the previous age due to individuals leaving the population before the analyzed age of TBI. As such, column % for ages are not provided and data on social determinants of health of the control cohort is based on the number of controls at age 0, the maximum number of controls in the study.

care physician visits from birth and each year after birth, up to 10 years post-injury, was significantly different by SDoH. For example, over the 10-year period post-injury, the rate of visits among males was higher than that of females until 7 to 9 years post-injury, after which they were similar (S2 Fig A); the rate among individuals living in urban neighbourhoods was higher than that of those living in rural neighbourhoods (S3 Fig A); and the rate among individuals living in areas with the most racialized and newcomer populations was higher than that of those in areas with the least racialized and newcomer populations (S4 Fig A).

## Rates of ED visits

Overall, rates of ED visits for the TBI cohort ranged from 33.2 per 100 person-years to 116.1 per 100 person-years and for the control cohort, from 21.3 to 71.0 per 100 person-years (Fig 1B). Rates for both the TBI cohort and controls were highest in the first two to three years of life, after which rates of ED visits gradually declined. Compared to controls, the rate of ED visits was significantly higher among the TBI cohort from birth, and in each year after birth, up to 10 years post-injury; specifically, the rate of ED visits among the TBI cohort was on average 29.0% (age 0) to 35.6% (age 4) higher than that of controls over the 10 years post-injury, when

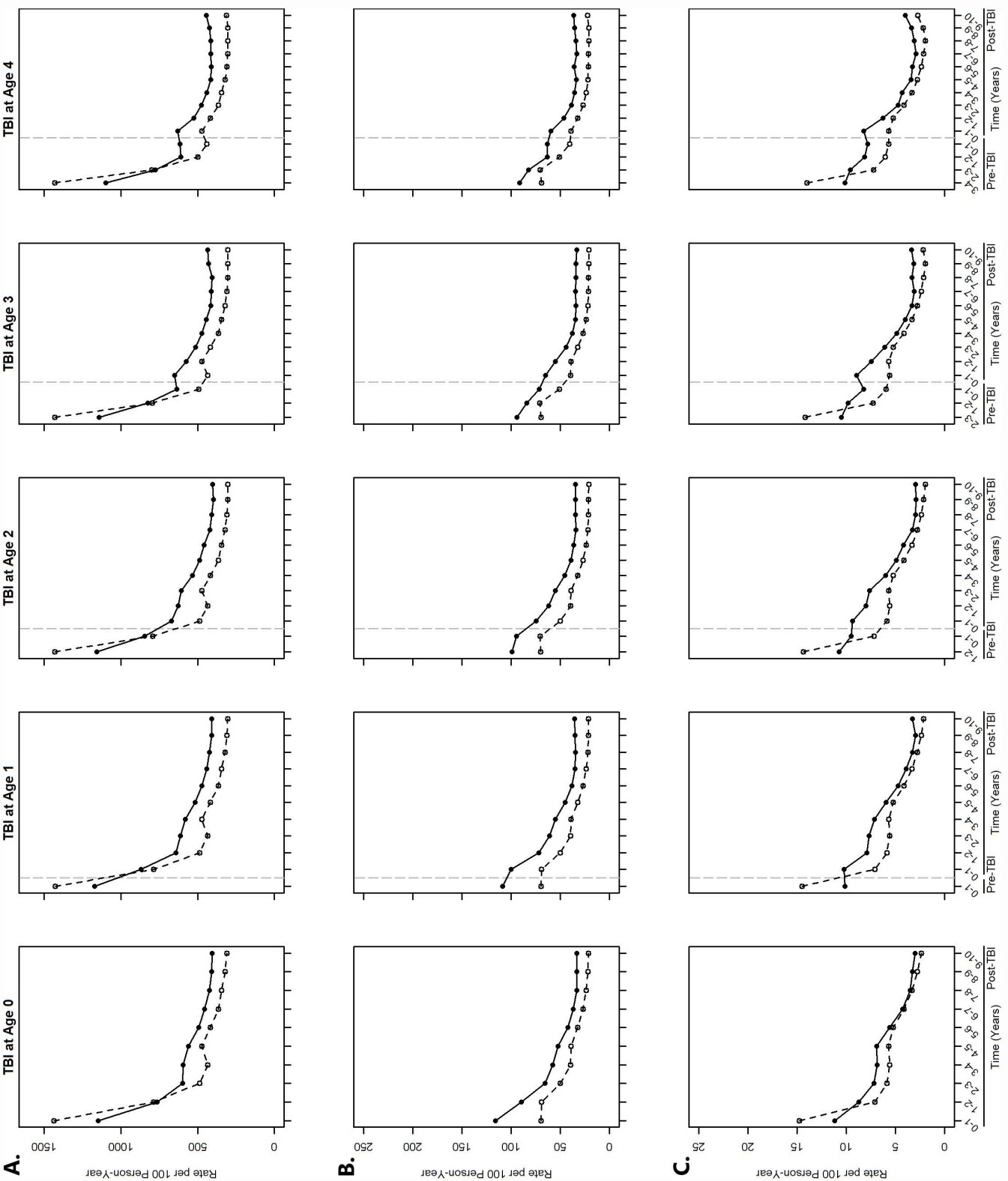

**Fig 1. Overall rates of healthcare use by age at incident TBI-related healthcare visit.** A. Primary care physician visits, B. Emergency Department visits, and C. hospitalizations. Solid black lines indicate the TBI cohort and dashed black lines indicate the controls. Dashed vertical grey lines indicate age 0 (i.e., birth).

stratified by age. Overall, the rate of ED visits from birth and in each year after birth, up to 10 years post-injury, was significantly different by SDoH. For example, over the 10-year period post-injury, the rate of visits among males was higher than that of females until 6 to 9 years post-injury, after which they were similar (S2 Fig B); the rate among individuals living in rural neighbourhoods was higher than that of those living in urban neighbourhoods (S3 Fig B); the rate among individuals living in the lowest income quintile was higher than that of individuals living in the highest income quintile, with the difference becoming smaller over time (S5 Fig B); and the rate among individuals living in areas with the least racialized and newcomer populations was higher than that of those in areas with the most racialized and newcomer populations (S4 Fig B).

## Rates of hospitalizations

Overall, rates of hospitalizations for the TBI cohort ranged from 2.9 per 100 person-years to 11.2 per 100 person-years and for the control cohort, from 2.0 to 14.8 per 100-person years (Fig 1C). Rates for both the TBI cohort and controls were highest in the first one to two years of life. Compared to controls, the rate of hospitalizations for TBI was significantly lower in the first year of life, after which they were significantly higher. The rate of hospitalizations among the TBI cohort was on average 9.1% (age 0) to 25.8% (age 4) higher than that of controls over the 10 years post-injury, when stratified by age. Overall, the rate of hospitalizations from birth and each year after birth, up to 10 years post-injury, was significantly different by SDoH. For example, over the 10-year period post-injury, the rate of visits among males was higher than that of females until 8 to 9 years post TBI for those experiencing a TBI at age 3 and 4, after which they were similar (S2 Fig C); the rate among individuals living in rural neighbourhoods was overall higher than that of those living in urban neighbourhoods (S3 Fig C); and the rate among individuals living in areas with the least racialized and newcomer populations was higher than that of those in areas with the most racialized and newcomer populations (S4 Fig C).

## Discussion

This is the first study, to the best of our knowledge, to follow children from birth up to 10 years post-TBI and compare their rates of healthcare use to children who did not have any TBI-related healthcare visits during the study period. Rates of healthcare use remained consistently higher in the TBI cohort compared to controls, up to 10 years post-injury. Rates of healthcare use were also higher prior to the first TBI-related healthcare visit, compared to controls, and differed across SDoH available in this birth cohort. Overall, preliminary analyses of data from this population-based birth cohort provide the foundation to catalyze in-depth research on longitudinal healthcare use to understand the causes of increased healthcare use over the 10-year period post-injury. It also provides evidence for integrating a lifespan perspective to understand healthcare use and needs after TBI and the need for research that takes into account SDoH.

Rates of healthcare use in the TBI cohort were higher than that of controls, both prior to the first TBI-related healthcare visit and in each year, up to 10 years post-TBI. While it is well-established that TBI is associated with increased healthcare use, this study showed that rates of healthcare use remained elevated even a decade after the first TBI-related healthcare visit. Research to understand the reasons for these elevated rates, particularly long-term post-TBI, is critical to identify opportunities for early intervention, especially of preventable hospitalizations and/or ED visits. Research that further explores similarities and differences in rates of healthcare use by injury severity is also encouraged. Over 95.0% of index TBI-related

healthcare visits in this birth cohort were from primary care physician visits, suggesting that these initial visits were for milder TBI (compared to visits to the hospital, which comprised of 3.8% of individuals in the TBI cohort). Economic evaluations of TBI have highlighted that while individual treatment costs of mild TBI are low, the high incidence of mild TBI results in "a total treatment cost across patients of nearly three times that for moderate-to-severe TBI" [4]. As such, research that stratifies longitudinal healthcare use by injury severity holds the potential to identify when and why individuals with TBI use primary care physician, ED, and hospital services. Finally, it is noteworthy that increases and decreases in rates of healthcare use among the TBI and control cohorts over time occur at similar timepoints in life, as reflected in the shape of the lines presented in Fig 1. This suggests that some healthcare use may be related to developmental stages of life, rather than the time of incident TBI-related healthcare visit. Nonetheless, rates of healthcare use in the TBI cohort remain higher compared to that of controls, further supporting the importance of exploring access to and use of healthcare over time.

Data from this preliminary analysis of longitudinal healthcare use also provides evidence that a lifespan perspective in research is critical. Even before the first TBI-related healthcare visit, rates of healthcare use in the TBI cohort were higher than that of those in controls; in fact, rates of ED visits were higher among the TBI cohort from birth. While it is possible that these higher rates of healthcare use may reflect visits related to the first TBI-related healthcare visit, a lifespan perspective to research can help us understand the impact of pre-existing health conditions and past health events on TBI, recovery, and subsequent healthcare needs. It also provides the opportunity for early intervention at the right time to possibly prevent a TBI or to facilitate early access to appropriate TBI care. This can set the foundation for lifelong wellness, as early childhood shapes later life.

Finally, this study showed that rates of healthcare use differed across SDoH. It is well-established that the consequences of TBI and SDoH often result in challenges in navigating the health system [14,15,19]. In this study, rates of ED visits were highest among those living in the lowest income quintile neighbourhoods, which may reflect a lack of access to primary care. The co-creation of research with individuals living in these neighbourhoods is critical to understand why rates of ED visits are high among those living in lowest income quintile neighbourhoods, and explore opportunities for targeted access to appropriate healthcare. Furthermore, future research, particularly those related to health service utilization, are encouraged to stratify data by SDoH to identify similarities and differences across SDoH to inform opportunities for targeted social care interventions.

## Strengths and limitations

We acknowledge the following limitations of this study. Only patients who sought and/or received medical care for a TBI were identified in the TBI cohort; those who did not seek medical care, did not receive medical care, or were not coded with a TBI diagnosis code would be misclassified into the control cohort. Furthermore, while this study showed changes in rates of healthcare use over time, the magnitude of the changes was not quantified; future research should implement further statistical analyses and modelling methods such as hazards ratios to understand the associations between TBI and healthcare use.

Nonetheless, this preliminary analysis of healthcare use is based on data from a population-based birth cohort that was built using health administrative data from a publicly funded health system where all residents receive medically necessary health services. As such, all healthcare use from birth and up to 10 years post-injury are reflected in this analysis, regardless of an individual's socioeconomic status. While this reduced sampling bias, we

acknowledge that rates of healthcare use presented in this analysis may not be generalizable to other countries with different health systems. Finally, this analysis is also the first, to the best of our knowledge, to present population-based and longitudinal data on healthcare use across age, sex, rurality of residence, neighbourhood income quintile, and neighbourhood racialized and newcomer populations characteristic, providing further support for the importance of research that accounts for SDoH.

## Conclusion

Preliminary analysis of longitudinal healthcare use from a population-based birth cohort showed that rates of healthcare use among the TBI cohort was consistently higher than that of controls, before the first TBI-related healthcare visit, and up to 10-years after this index visit, and differed by SDoH. These findings highlight the need for further research to understand why the rates of healthcare use among the TBI cohort remain elevated compared to controls, over a 10-year period. In addition, this analysis provides evidence supporting a lifespan perspective to understanding healthcare use and needs after TBI, as rates of healthcare use were higher among the TBI cohort even before the index TBI-related healthcare visit. Finally, rates of healthcare use differed by SDoH, highlighting the importance of research that is stratified by SDoH to identify opportunities for targeted social care supports in accessing healthcare.

## Supporting information

**S1 Table. International classification of diseases and related health problems version 10 and ontario health insurance plan codes used to identify traumatic brain injury-related healthcare visits.**
(PDF)

**S1 Fig. Flow diagram of study sample.**
(TIF)

**S2 Fig. Rate of healthcare by sex and age at incident TBI-related healthcare visit:** A. Primary care physician visits, B. ED visits, and C. hospitalizations. Dashed vertical lines indicate age 0 (i.e., birth).
(ZIP)

**S3 Fig. Rates of healthcare use by rurality of residence and age at incident TBI-related healthcare visit:** A. Primary care physician visits, B. ED visits, and C. hospitalizations. Dashed vertical lines indicate age 0 (i.e., birth).
(ZIP)

**S4 Fig. Rates of healthcare use by neighbourhood racialized and newcomer populations characteristic and age at incident TBI-related healthcare visit:** A. Primary care physician visits, B. ED visits, and C. hospitalizations. Dashed vertical lines indicate age 0 (i.e., birth).
(ZIP)

**S5 Fig. Rates of healthcare use by neighbourhood income quintile and age at incident TBI-related healthcare visit:** A. Primary care physician visits, B. ED visits, and C. hospitalizations. Dashed vertical lines indicate age 0 (i.e., birth).
(ZIP)

**S1 Text. Explanation of computation limitations.**
(PDF)

## Author contributions

**Conceptualization:** Vincy Chan, Robert Balogh, Michael David Escobar.

**Data curation:** Vincy Chan.

**Formal analysis:** Vincy Chan, Clarissa Serafine Wirianto, Michael David Escobar.

**Funding acquisition:** Vincy Chan.

**Investigation:** Vincy Chan.

**Methodology:** Vincy Chan, Clarissa Serafine Wirianto, Michael David Escobar.

**Project administration:** Vincy Chan.

**Resources:** Vincy Chan.

**Supervision:** Vincy Chan, Michael David Escobar.

**Validation:** Vincy Chan, Michael David Escobar.

**Visualization:** Vincy Chan, Clarissa Serafine Wirianto, Michael David Escobar.

**Writing – original draft:** Vincy Chan.

**Writing – review & editing:** Vincy Chan, Clarissa Serafine Wirianto, Robert Balogh, Michael David Escobar.

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
