## [Decision Letter · Decision Letter 0]

14 Nov 2024

PONE-D-24-35301Longitudinal healthcare use after pediatric brain injury: A population-based birth cohort studyPLOS ONE

Dear Dr. Chan,

Thank you for submitting your manuscript to PLOS ONE. After careful consideration, we feel that it has merit but does not fully meet PLOS ONE’s publication criteria as it currently stands. Therefore, we invite you to submit a revised version of the manuscript that addresses the points raised during the review process.

We look forward to receiving your revised manuscript.

Kind regards,

Masaki Mogi

Academic Editor

PLOS ONE

**Journal Requirements:**

Research reported in this publication was supported by the Eunice Kennedy Shriver National Institute of Child Health & Human Development of the National Institutes of Health under Award Number R03HD104206. The funders had no role in study design, data collection and analysis, decision to publish, or preparation of the manuscript. The content is solely the responsibility of the authors and does not necessarily represent the views of the National Institutes of Health.

URL: https://www.nichd.nih.gov/grants-contracts

This study contracted ICES Data & Analytic Sciences (DAS) and used de 314 identified data from the ICES Data Repository, which is managed by ICES with support from its funders and partners: Canada’s Strategy for Patient-Oriented Research (SPOR), the Ontario SPOR Support Unit, the Canadian Institutes of Health Research and the Government of Ontario. The opinions, results and conclusion reported are those of the authors.

Research reported in this publication was supported by the Eunice Kennedy Shriver National Institute of Child Health & Human Development of the National Institutes of Health under Award Number R03HD104206. The funders had no role in study design, data collection and analysis, decision to publish, or preparation of the manuscript. The content is solely the responsibility of the authors and does not necessarily represent the views of the National Institutes of Health.

URL: https://www.nichd.nih.gov/grants-contracts

5. Please note that your Data Availability Statement is currently missing the direct link to access each database. If your manuscript is accepted for publication, you will be asked to provide these details on a very short timeline. We therefore suggest that you provide this information now, though we will not hold up the peer review process if you are unable.

6. We notice that your supplementary table are included in the manuscript file. Please remove them and upload them with the file type 'Supporting Information'. Please ensure that each Supporting Information file has a legend listed in the manuscript after the references list.. 

Reviewers' comments:

Reviewer's Responses to Questions

**Comments to the Author**

1. Is the manuscript technically sound, and do the data support the conclusions?

Reviewer #1: Yes

2. Has the statistical analysis been performed appropriately and rigorously? 

Reviewer #1: Yes

3. Have the authors made all data underlying the findings in their manuscript fully available?

Reviewer #1: Yes

4. Is the manuscript presented in an intelligible fashion and written in standard English?

Reviewer #1: Yes

5. Review Comments to the Author

**Reviewer #1: ** This study examines long-term healthcare use in children (ages 0-4) with traumatic brain injury (TBI) compared to controls. Findings show that TBI-affected children consistently had higher rates of primary care visits, emergency department visits, and hospitalizations. Healthcare use varied by gender, urban/rural location, and socioeconomic factors.

While the study presents valuable insights, some points would benefit from further refinement:

1. Some clarification about the random selection approach used to choose 10% of the sample groups is necessary. Please report the specific method utilized for this random selection. Given the potential for variability, different sampling methods or alternative samples could produce divergent results. Future analyses should replicate these results using varied sample selections to improve the validity of the findings.

2. Table 1 could also help to determine whether there are statistical differences between the two groups regarding the sociodemographic variables. Please include some group comparison statistics.

3. It is unclear why specific exclusion criteria, such as comorbid conditions (e.g., epilepsy, autism), are absent. Including these criteria that could significantly influence healthcare utilization would help isolate the specific impact of TBI.

4. While the introduction provides an adequate overview of the study’s context, more information on the neurological impacts of TBI would be useful. In particular, a more comprehensive understanding of why such injuries lead to increased healthcare utilization and the specific challenges this population faces would be beneficial.

5. Consider providing a supplementary table summarizing the healthcare usage rates presented in the results section to make them more accessible for those who prefer data visualization over narrative explanations.

6. In the statistical analysis section, including a mathematical formula used for the calculations would be beneficial.

7. The quality of the figures could be improved, specifically in the resolution.

8. It would be helpful to include more information about the software (e.g., R, Python, SPSS) used for statistical analysis and the computational resources employed during the study.

6. PLOS authors have the option to publish the peer review history of their article (what does this mean? ). If published, this will include your full peer review and any attached files.

**Do you want your identity to be public for this peer review?** For information about this choice, including consent withdrawal, please see our Privacy Policy .

Reviewer #1: **Yes: ** Emiliano Trimarco

---

## [Author Response · Author response to Decision Letter 0]

27 Nov 2024

Responses to Reviewers’ Comments

We would like to thank the editor and reviewers for their time and feedback for manuscript. We have addressed your comments below in blue font. Page and line numbers correspond to the simple markup version without tracked changes.

Thank you again for your time. We would be happy to address any other comments that arise from a review of the new content.

Reviewer #1

This study examines long-term healthcare use in children (ages 0-4) with traumatic brain injury (TBI) compared to controls. Findings show that TBI-affected children consistently had higher rates of primary care visits, emergency department visits, and hospitalizations. Healthcare use varied by gender, urban/rural location, and socioeconomic factors.

While the study presents valuable insights, some points would benefit from further refinement:

1. Some clarification about the random selection approach used to choose 10% of the sample groups is necessary. Please report the specific method utilized for this random selection. Given the potential for variability, different sampling methods or alternative samples could produce divergent results. Future analyses should replicate these results using varied sample selections to improve the validity of the findings.

We added clarification on the 10% random selection of samples, for which we used a uniform distribution. Please see page 6, line 112.

2. Table 1 could also help to determine whether there are statistical differences between the two groups regarding the sociodemographic variables. Please include some group comparison statistics.

We added group comparison statistics in the form of weighted Chi-squared tests between the cohort and each characteristic variable. Please see page 7, lines 131-132 and pages 8-9, Table 1.

3. It is unclear why specific exclusion criteria, such as comorbid conditions (e.g., epilepsy, autism), are absent. Including these criteria that could significantly influence healthcare utilization would help isolate the specific impact of TBI.

Thank you for this comment. To date, there is limited longitudinal data that follow children with TBI. The goal of this study is to address this gap and provide foundational data on longitudinal healthcare use. As such, this study explored and compared the overall rates of healthcare use between individuals with an early-life TBI compared to those without an early-life TBI. Future research will include identifying the specific health conditions that contribute to increased healthcare use.

4. While the introduction provides an adequate overview of the study’s context, more information on the neurological impacts of TBI would be useful. In particular, a more comprehensive understanding of why such injuries lead to increased healthcare utilization and the specific challenges this population faces would be beneficial.

Supporting literature on the neurological impact of TBIs and their association to increased healthcare utilization is added to the introduction section. Please see page 4, lines 59-63.

5. Consider providing a supplementary table summarizing the healthcare usage rates presented in the results section to make them more accessible for those who prefer data visualization over narrative explanations.

We have provided these tables in the supporting information section. Please see S2 Tables.

6. In the statistical analysis section, including a mathematical formula used for the calculations would be beneficial.

We added mathematical formulas for rate and confidence interval calculations to the methods section. Please see page 8, lines 140 and 147.

7. The quality of the figures could be improved, specifically in the resolution.

We have improved the resolution of the images to 300 DPI and uploaded them as a separate file.

8. It would be helpful to include more information about the software (e.g., R, Python, SPSS) used for statistical analysis and the computational resources employed during the study.

We have added the specific computational resources and programs used for analysis to the methods section, principally R and parallel processing. Please see page 7, lines 129-130.

Editor

We have updated the manuscript and other files to reflect the PLOS ONE style requirements.

We’ve provided the tracked changes in word format. Please let us know if LaTeX is absolutely required. the LaTeX version did not allow for tracked changes.

Research reported in this publication was supported by the Eunice Kennedy Shriver National Institute of Child Health & Human Development of the National Institutes of Health under Award Number R03HD104206. The funders had no role in study design, data collection and analysis, decision to publish, or preparation of the manuscript. The content is solely the responsibility of the authors and does not necessarily represent the views of the National Institutes of Health.

URL: https://www.nichd.nih.gov/grants-contracts

We have amended our Funding Statement to clarify all sources of funding. We have also included this amended statement in the cover letter. Please see page 17, lines 302-306 of the manuscript.

This study contracted ICES Data & Analytic Sciences (DAS) and used de 314 identified data from the ICES Data Repository, which is managed by ICES with support from its funders and partners: Canada’s Strategy for Patient-Oriented Research (SPOR), the Ontario SPOR Support Unit, the Canadian Institutes of Health Research and the Government of Ontario. The opinions, results and conclusion reported are those of the authors.

Research reported in this publication was supported by the Eunice Kennedy Shriver National Institute of Child Health & Human Development of the National Institutes of Health under Award Number R03HD104206. The funders had no role in study design, data collection and analysis, decision to publish, or preparation of the manuscript. The content is solely the responsibility of the authors and does not necessarily represent the views of the National Institutes of Health.

URL: https://www.nichd.nih.gov/grants-contracts

We have removed funding information from the acknowledgements. Please see page 17, lines 307-308.

5. Please note that your Data Availability Statement is currently missing the direct link to access each database. If your manuscript is accepted for publication, you will be asked to provide these details on a very short timeline. We therefore suggest that you provide this information now, though we will not hold up the peer review process if you are unable.

Our Data Availability Statement notes that data sharing agreements prohibit the dataset from becoming publicly available. We cannot provide a direct link to the dataset.

6. We notice that your supplementary table are included in the manuscript file. Please remove them and upload them with the file type 'Supporting Information'. Please ensure that each Supporting Information file has a legend listed in the manuscript after the references list.

We have removed the Supporting Information table, images, and text, and uploaded them separately. We have done the same for Figures in the main body.

We hope that these revisions have addressed the Editor’s and Reviewer’s concerns. Once again, we sincerely thank you for your time and efforts in improving our manuscript, and we look forward to your response.

---

## [Decision Letter · Decision Letter 1]

9 Dec 2024

Longitudinal healthcare use after pediatric brain injury: A population-based birth cohort study

PONE-D-24-35301R1

Dear Dr. Chan,

We’re pleased to inform you that your manuscript has been judged scientifically suitable for publication and will be formally accepted for publication once it meets all outstanding technical requirements.

Kind regards,

Masaki Mogi

Academic Editor

PLOS ONE

Additional Editor Comments (optional):

Reviewers' comments:

Reviewer's Responses to Questions

**Comments to the Author**

1. If the authors have adequately addressed your comments raised in a previous round of review and you feel that this manuscript is now acceptable for publication, you may indicate that here to bypass the “Comments to the Author” section, enter your conflict of interest statement in the “Confidential to Editor” section, and submit your "Accept" recommendation.

Reviewer #1: All comments have been addressed

2. Is the manuscript technically sound, and do the data support the conclusions?

Reviewer #1: Yes

3. Has the statistical analysis been performed appropriately and rigorously? 

Reviewer #1: Yes

4. Have the authors made all data underlying the findings in their manuscript fully available?

Reviewer #1: No

5. Is the manuscript presented in an intelligible fashion and written in standard English?

Reviewer #1: Yes

6. Review Comments to the Author

Reviewer #1: (No Response)

7. PLOS authors have the option to publish the peer review history of their article (what does this mean? ). If published, this will include your full peer review and any attached files.

**Do you want your identity to be public for this peer review?** For information about this choice, including consent withdrawal, please see our Privacy Policy .

Reviewer #1: **Yes: ** Trimarco Emiliano

---

## [Editor Report · Acceptance letter]

PONE-D-24-35301R1

PLOS ONE

Dear Dr. Chan,

I'm pleased to inform you that your manuscript has been deemed suitable for publication in PLOS ONE. Congratulations! Your manuscript is now being handed over to our production team.

Kind regards,

on behalf of

Dr. Masaki Mogi

Academic Editor

PLOS ONE